# Efficient Reinforcement Learning in Resource Allocation Problems Through Permutation Invariant Multi-task Learning

## Abstract

One of the main challenges in real-world reinforcement learning is to learn successfully from limited training samples. We show that in certain settings, the available data can be dramatically increased through a form of multi-task learning, by exploiting an invariance property in the tasks. We provide a theoretical performance bound for the gain in sample efficiency under this setting. This motivates a new approach to multi-task learning, which involves the design of an appropriate neural network architecture and a prioritized task-sampling strategy. We demonstrate empirically the effectiveness of the proposed approach on two real-world sequential resource allocation tasks where this invariance property occurs: financial portfolio optimization and meta federated learning.

## 1 Introduction

Sample efficiency in reinforcement learning (RL) is an elusive goal. Recent attempts at increasing the sample efficiency of RL implementations have focused to a large extent on incorporating models into the training process: Xu et al. (2019); Clavera et al. (2018); Zhang et al. (2018); Berkenkamp et al. (2017); Ke et al. (2019); Yarats et al. (2019); Huang et al. (2019); Chua et al. (2018); Serban et al. (2018). The models encapsulate knowledge explicitly, complementing the experiences that are gained by sampling from the RL environment. Another means towards increasing the availability of samples for a reinforcement learner is by tilting the training towards one that will better transfer to related tasks: if the training process is sufficiently well adapted to more than one task, then the training of a particular task should be able to benefit from samples from the other related tasks. This idea was explored a decade ago in Lazaric & Ghavamzadeh (2010) and has been gaining traction ever since, as researchers try to increase the reach of deep reinforcement learning from its comfortable footing in solving games outrageously well to solving other important problems. Yu (2018) discusses a number of methods for increasing sample efficiency in RL and includes experience transfer as one important avenue, covering the transfer of samples, as we do here, transfer of representation or skills, and jumpstarting models which are then ready to be quickly, i.e. with few samples, updated to different tasks. D'Eramo et al. (2020) address the same idea, noting that multi-task learning can improve the learning of each individual task, motivated by robotics-type tasks with underlying commonality, such as balancing a single vs. a double pendulum, or hopping vs. walking.

We are interested in exploiting the ability of multi-task learning to solve the sample efficiency problem of RL. Our setting does not apply to all problem classes nor does it seek to exploit the kind of physical similarities found in robotics tasks that form the motivation of Lazaric & Ghavamzadeh (2010); D'Eramo et al. (2020). Rather, we show that there are a number of reinforcement learning tasks with a particular fundamental property that makes them ideal candidates for multi-task learning with the goal of increasing the availability of samples for their training. We refer to this property as permutation invariance. It is present in very diverse tasks: we illustrate it on a financial portfolio optimization problem, whereby trades are executed sequentially over a given time horizon, and on the problem of meta-learning in a federated supervised learning setting.

Permutation invariance in the financial portfolio problem exhibits itself as follows: consider the task of allocating a portion of wealth to each of a number of financial instruments using a trading policy. If the trading policy is permutation invariant, one can change the order of the instruments without

changing the policy. This allows one to generate multiple portfolio optimization tasks from a given set of financial instruments. A commonality between applications that have this property is that they concern sequential resource allocation: at each time step, the resource allocation scores the quality of each available candidate entity (for example a financial instrument in the above example), then based on those scores, apportions out the resource (the total wealth to invest, in the above example) among the entities at that time step, so that over the horizon of interest, the reward is maximized.

Sequential resource allocation problems include applications such as sequential allocation of budget, sequential allocation of space, e.g. in IT systems, hotels, delivery vehicles, sequential allocation of people to work slots or appointments, etc. Many such applications possess permutation invariance in that the ordering of the entities, i.e. where the resources are allocated, can change without changing the resulting optimal allocation. We show that under this form of permutation invariance, it is possible to derive a bound on the performance of the policy. The bound is an extension of that of Lazaric & Ghavamzadeh (2010), and while similar to, provides additional information beyond the bound of D'Eramo et al. (2020). We use the bound to motivate an algorithm that allows for substantially improved results as compared with solving each task on its own. The bound and the algorithm are first analyzed on a synthetic problem that validates the bound in our theorem and confirms the multi-task gain that the theory predicts. Hessel et al. (2018); Bram et al. (2019) have cautioned against degrading of the performance on each task when some tasks bias the updates to the detriment of others in multi-task learning. They claim that some tasks have a greater density or magnitude of in-task rewards and hence a disproportionate impact on the learning process. In our setting, deleterious effects of some tasks on others could also arise. The algorithm we propose handles this through a form of prioritized sampling, where priorities are put on the tasks themselves, and acts like a prioritized experience replay buffer, applied to a multi-task learning problem. We show empirically that the priorities thus defined protect the overall learning problem from the deleterious effects that unrelated or unhelpful tasks could otherwise have on the policy.

The contributions of this work are as follows: (1) we identify the permutation invariance property of the class of reinforcement learning problems involving sequential resource allocation, (2) we define a method to increase sample efficiency in these reinforcement learning problems by leveraging this property of permutation invariance; (3) we provide a theoretical performance bound for the class of problems; (4) we validate experimentally the utility of permutation variance on sample efficiency as well as the validity of the bound on a synthetic problem; and (5) we illustrate two real-world RL resource allocation tasks for which this property holds and demonstrate the benefits of the proposed method on sample efficiency and thus also on the overall performance of the models.

## 2 RELATED WORK

A notable first stream of work on leveraging multi-task learning for enhancing RL performance on single tasks can be found in Wilson et al. (2007); Lazaric & Ghavamzadeh (2010) which consider, as we do, that there is an underlying MDP from which the multiple tasks can be thought to derive. They use however a Bayesian approach and propose a different algorithmic method than ours. Our results extend performance bounds by Lazaric et al. (2012) on single-task RL. As noted by Yu (2018), jumpstarting, or distilling experiences and representations of relevant policies is another means to increasing sample efficiency in solving a new but related problem. Rusu et al. (2016) uses this idea in so-called progressive neural networks and Parisotto et al. (2015) leverage multiple experts to guide the derivation of a general policy. With a similar objective, Teh et al. (2017) define a policy centroid, that is, a shared distilled policy, that captures the commonalities across the behaviors in the tasks. In all of these distillation-type methods, the tasks considered are simple or complex games.

Teh et al. (2017) note that their policy centroid method, distral, is likely to be affected by task interference, in that differences across tasks may degrade the performance of the resulting policy of any of the constituent tasks. This topic was studied by Hessel et al. (2018); Bram et al. (2019). Hessel et al. (2018) proposed a solution to this by extending the so-called PopArt normalization van Hasselt et al. (2016) to re-scale the updates of each task so that the different characteristics of the task-specific reward do not skew the learning process. Bram et al. (2019) use a different approach that learns attention weights of the sub-networks of each task and discards those that are not relevant or helpful. Vuong et al. (2019); D'Eramo et al. (2020) are, like our work, concerned with sharing of experiences to facilitate a more sample-efficient learning process. Vuong et al. (2019) suggest

identifying the shared portions of tasks to allow sharing of samples in those portions. The work of D'Eramo et al. (2020) is in some ways quite similar to ours: the authors' goal is the same and they derive a bound as we do on the performance in this setting. However, their setting is different in that their tasks have both shared and task-specific components, and their bound becomes tighter only as the number of tasks increases. In our setting, we do not require a task-specific component, and we are able to show how the distance between the MDPs of each task, in addition to the number of tasks, affects the strength of the bound. Recently, permutation invariance has been exploited in deep multi-agent reinforcement learning (Liu et al., 2019) where the invariance properties arise naturally in a homogeneous multi-agent setting. Their work employs permutation invariance in learning the critic whereas in our case the entire learned policy employs permutation invariance.

## 3 PRELIMINARIES

We begin by defining notation. For a measurable space with domain $\mathcal{X}$, let $\mathcal{S}(\mathcal{X})$ denote the set of probability measures over $\mathcal{X}$, and $\mathcal{B}(\mathcal{X}; L)$ the space of bounded measurable functions with domain $\mathcal{X}$ and bound $0 < L < \infty$. For a measure $\rho \in \mathcal{S}(\mathcal{X})$ and a measurable function $f : \mathcal{X} \to \mathbb{R}$, the $l_2(\rho)$-norm of $f$ is $\|f\|_\rho$, and for a set of $n$ points $X_1, \cdots, X_n \in \mathcal{X}$, the empirical norm, $\|f\|_n$ is

$$\|f\|_\rho^2 = \int f(x)^2 \rho(dx) \quad \text{and} \quad \|f\|_n^2 = \frac{1}{n} \sum_{t=1}^n f(X_t)^2.$$

Let $\|f\|_\infty = \sup_{x \in \mathcal{X}} |f(x)|$ be the supremum norm of $f$. Consider a set of MDPs indexed by $t$. Each MDP is denoted by a tuple $\mathcal{M}_t = \langle \mathcal{X}, \mathcal{A}, R_t, P_t, \gamma \rangle$, where $\mathcal{X}$, a bounded closed subset of the $s$-dimensional Euclidean space, is a common state space; $\mathcal{A}$ is a common action space, $R_t : \mathcal{X} \times \mathcal{A} \to \mathbb{R}$ is a task specific reward function uniformly bounded by $R_{\max}$, $P_t$ is a task specific transition kernel such that $P_t(\cdot|x, a)$ is a distribution over $\mathcal{X}$ for all $x \in \mathcal{X}$ and $a \in \mathcal{A}$, and $\gamma \in (0, 1)$ is a common discount factor. Deterministic policies are denoted by $\pi : \mathcal{X} \to \mathcal{A}$. For a given policy $\pi$, the MDP $\mathcal{M}_t$ is reduced to a Markov chain $\mathcal{M}_t^\pi = \langle \mathcal{X}, R_t^\pi, P_t^\pi, \gamma \rangle$ with reward function $R_t^\pi(x) = R_t(x, \pi(x))$, transition kernel $P_t^\pi(\cdot|x) = P_t(\cdot|x, \pi(x))$, and stationary distribution $\rho_t^\pi$. The value function $V_t^\pi$ for MDP $t$ is defined as the unique fixed-point of the Bellman operator $\mathcal{T}_t^\pi : \mathcal{B}(\mathcal{X}; V_{\max} = R_{\max}/(1 - \gamma)) \to \mathcal{B}(\mathcal{X}; V_{\max})$, given by

$$(\mathcal{T}_t^\pi V)(x) = R_t^\pi(x) + \gamma \int_{\mathcal{X}} P_t^\pi(dy|x) V(y).$$

Let $\pi_t^*$ denote the optimal policy for $\mathcal{M}_t$. The optimal value function $V_t^{\pi_t^*}$ for $\mathcal{M}_t$ is defined as the unique fixed-point of its optimal Bellman operator $\mathcal{T}_t^{\pi_t^*}$ which is defined by

$$(\mathcal{T}_t^{\pi_t^*} V)(x) = \max_{a \in \mathcal{A}} \left[ R_t(x, a) + \gamma \int_{\mathcal{X}} P_t(dy|x, a) V(y) \right].$$

To approximate the value function $V$, we use a linear approximation architecture with parameters $\alpha \in \mathbb{R}^d$ and basis functions $\varphi_i \in \mathcal{B}(\mathcal{X}; L)$ for $i = 1, \cdots, d$. Let $\varphi(\cdot) = (\varphi_1(\cdot), \cdots, \varphi_d(\cdot))^\mathsf{T} \in \mathbb{R}^d$ be the feature vector and $\mathcal{F}$ the linear function space spanned by basis functions $\varphi_i$. Thus, $\mathcal{F} = \{f_\alpha \mid \alpha \in \mathbb{R}^d \text{ and } f_\alpha(\cdot) = \varphi(\cdot)^\mathsf{T} \alpha\}$.

Consider a learning task to dynamically allocate a common resource across entities $\mathcal{U}_t \subseteq \mathcal{U}$. Each $t$ corresponds to a task, but for now take $t$ to be an arbitrary fixed index. At each time step $n$, the decision maker observes states $x_n = (x_{i,n})_{i \in \mathcal{U}_t}$ of the entities, where $x_{i,n}$ is the state of entity $i$, and takes action $a_n = (a_{i,n})_{i \in \mathcal{U}_t}$, where $a_{i,n}$ is the share of the resource allocated to entity $i$. The total resource capacity is normalized to 1 for convenience. Therefore, allocations satisfy $0 \le a_{i,n} \le 1$ and $\sum_{i \in \mathcal{U}_t} a_{i,n} = 1$. We consider policy $\pi_\theta(x_n)$ parameterized by $\theta$. Assume that we have access to the reward function $R_t$ as well as a simulator that generates a trajectory of length $N$ given any arbitrary policy $\pi_\theta$. The objective of the learning task is to maximize

$$J_t(\theta) = \mathbb{E} \left[ \sum_{n=1}^N \gamma^{n-1} R_t(x_n, a_n) \,\middle|\, a_{n+1} = \pi_\theta(x_n), \; x_{n+1} \sim P_t(\cdot|x_n, a_n), \; x_1 \sim P_t(\cdot) \right].$$

In many settings, $N$ is small and simulators are inaccurate; therefore, trajectories generated by the simulator are poor representations of the actual transition dynamics. This occurs in batch RL where trajectories are rollouts from a dataset. In these cases, policies overfit and generalize poorly.

## 4 THEORETICAL RESULTS

We introduce first a property that we term permutation-invariance for the policy network that can be shown to help significantly reduce overfitting.

**Definition 1 (Permutation Invariant Policy Network)** *A policy network $\pi_\theta$ is permutation invariant if it satisfies $\pi_\theta(\sigma(x)) = \sigma(\pi_\theta(x))$ for any permutation $\sigma$.*

Permutation invariant policy networks have significant advantages over completely integrated policy networks. While the latter are likely to fit correlations between different entities, this is not possible with permutation invariant policy networks as they are agnostic to identities of entities. Therefore, permutation invariant policy networks are better able to leverage experience across time and entities, leading to greater efficiency in data usage. Moreover, observe that if the transition kernels can be factored into independent and identical transition kernels across entities, then the optimal policy is indeed permutation invariant.

Our main theoretical contributions start with an extension of results from Lazaric et al. (2012), where a finite-sample error bound was derived for the least squares policy iteration (LSPI) algorithm on a single task. Lazaric et al. (2012) provided a high-probability bound on the performance difference between the final learned policy and the optimal policy, of the form $c_1 + c_2/\sqrt{N}$, where $c_1$ and $c_2$ are constants that depend on the task and the chosen feature space, and $N$ is the number of training examples. We extend their result by showing that, as long as tasks are $\epsilon$-close to each other (with respect to a similarity measure we define later), the error bound of solving each task using our multi-task approach has the form $c_1 + c_2/\sqrt{NT} + c_3\epsilon$, where $T$ is the number of tasks and $c_3$ is a task-dependent constant. Specifically, our theorem provides a general result and performance guarantee with respect to using data from a different but similar MDP. Definition 1 provides a basis for generating many such MDPs. Finally, the benefit of doing so shall be provided by Corollary 2. Thus, provided $\epsilon$ is small, a given task can benefit from a much larger set of $NT$ training examples.

In addition to the assumptions of Lazaric et al. (2012), we extend the definition of second-order discounted-average concentrability, proposed in Antos et al. (2008), and define the notion of first-order discounted-average concentrability. The latter will be used in our main result, Theorem 1.

**Assumption 1** *There exists a distribution $\mu \in \mathcal{S}(\mathcal{X})$ such that for any policy $\pi$ that is greedy with respect to a function in the truncated space $\tilde{\mathcal{F}}$, $\mu \le C\rho_t^\pi$ for all $t$, where $C < \infty$ is a constant. Given the target distribution $\sigma \in \mathcal{S}(\mathcal{X})$ and an arbitrary sequence of policies $\{\pi_m\}_{m\ge 1}$, let*

$$c_{\sigma,\mu} = \sup_{\pi_1,\dots,\pi_m} \left\| \frac{d(\mu P^{\pi_1} \dots P^{\pi_m})}{d\sigma} \right\|.$$

*We assume that $C'_{\sigma,\mu}, C''_{\sigma,\mu} < \infty$, and define first and second order discounted-average concentrability of future-state distributions as follows:*

$$C'_{\sigma,\mu} = (1-\gamma) \sum_{m\ge 0} \gamma^m c_{\sigma,\mu}(m),$$

$$C''_{\sigma,\mu} = (1-\gamma)^2 \sum_{m\ge 1} m\gamma^{m-1} c_{\sigma,\mu}(m).$$

**Theorem 1 (Multi-Task Finite-Sample Error Bound)** *Let $\mathcal{M} = \langle \mathcal{X}, \mathcal{A}, R, P, \gamma \rangle$ be an MDP with reward function $R$ and transition kernel $P$. Assume $\mathcal{A}$ finite. Denote its Bellman operator by*

$$(\mathcal{T}^\pi V)(x) = R^\pi(x) + \gamma \int_{\mathcal{X}} P^\pi(dy|x)V(y).$$

*Given a policy $\pi$, define the Bellman difference operator between $\mathcal{M}_t$ and $\mathcal{M}$ to be $\mathcal{D}_t^\pi V = \mathcal{T}_t^\pi V - \mathcal{T}^\pi V$. Apply the LSPI algorithm to $\mathcal{M}$, by generating, at each iteration $k$, a path from $\mathcal{M}$ of size $N$, where $N$ satisfies Lemma 4 in Lazaric et al. (2012). Let $V_{-1} \in \tilde{\mathcal{F}}$ be an arbitrary initial value function, $V_0, \dots, V_{K-1}$ $(\tilde{V}_0, \dots, \tilde{V}_{K-1})$ be the sequence of value functions (truncated value functions) generated by the LSPI after $K$ iterations, and $\pi_k$ be the greedy policy w.r.t. the truncated value function $\tilde{V}_{k-1}$. Suppose also that*

$$\|\mathcal{D}_t^\pi V^\pi\|_\mu \le \epsilon \ \forall \ \pi, \quad and \quad \|\mathcal{D}_t^{\pi_k} \tilde{V}_{k-1}\|_\mu \le \epsilon \ \forall \ k.$$

*Then, for constants $c_1$, $c_2$, $c_3$, $c_4$ that are dependent on $\mathcal{M}$, with probability $1 - \delta$ (with respect to the random samples):*

$$\|V_t^{\pi_t^*} - V_t^{\pi_K}\|_\sigma \leq c_1 \frac{1}{\sqrt{N}} + c_2 \epsilon \sqrt{C'_{\sigma,\mu}} + c_3 \sqrt{C''_{\sigma,\mu}} + c_4.$$

The proof is deferred to the Appendix. Theorem 1 formalizes the trade off between drawing fewer samples from the exact MDP $\mathcal{M}_t$, versus drawing more samples from a different MDP $\mathcal{M}$. Importantly, it shows how to benefit from solving a different MDP, $\mathcal{M}$, when: (a) additional samples can be obtained from $\mathcal{M}$, and (b) $\mathcal{M}$ is not too different from $\mathcal{M}_t$. In particular, the distance measure is simply the distance between the Bellman operators of the MDPs, which can be bounded if the difference in both the transition and reward functions are bounded.

In recent work, a performance bound for multi-task learning was given in Theorem 2 and 3 of D'Eramo et al. (2020). However, the authors used a different setup containing both shared and task-specific representations, and their focus was on showing that the cost of learning the shared representation decreases with more tasks. They did not show how the similarity or difference across tasks affects performance. In contrast, our setup does not contain task-specific representations, and our focus is on how differences across MDPs impact the benefit of having more tasks (and consequently more samples). We show this in Corollary 1 and Corollary 2.

**Remark 1** *While our theoretical results are based on LSTD and LSPI and assume finite action space, our approach is applicable to a wide range of reinforcement learning algorithms, including policy gradient methods and to MDPs with continuous action spaces. Deriving similar results for a larger family of models and algorithms remains an interesting, albeit challenging, future work.*

Permutation invariant policy networks allow using data from the global set of entities $\mathcal{U}$. Since the policy network is agnostic to the identities of the entities, one can learn a single policy for all tasks, where each task $t \in [T]$ is a resource allocation problem over a subset of entities $\mathcal{U}_t$. For notational simplicity, assume that all tasks have the same number of entities, and all trajectories are of equal length $N$. Our approach can, however, be readily extended to tasks with different numbers of entities and different trajectory lengths. Permutation invariance allows a large set of MDPs to leverage the result of Theorem 1. In the next section we shall provide an algorithm, motivated by the following corollaries, and a prioritized sampling strategy for this setting that drives significantly greater sample efficiency for the original task. The sampling strategy also helps to stabilize the learning process, reducing the risk of deleterious effects of the multi-task setting, as discussed by Teh et al. (2017) and addressed in works such as Hessel et al. (2018); Bram et al. (2019).

**Corollary 1** *Let $[T]$ be a set of similar tasks such that their distance from the average MDP, given by*

$$(\mathcal{T}^\pi V)(x) = \frac{1}{T} \sum_{t=1}^T R_t^\pi(x) + \gamma \int_\mathcal{X} \frac{1}{T} \sum_{t=1}^T P_t^\pi(dy|x)V(y),$$

*is bounded by $\epsilon$ as defined in Theorem 1. Let $N$ be the number of samples available in each task. Let $\pi_K$ be the policy obtained at the $K^{th}$ iteration when applying LSPI to the average MDP. Then, the suboptimality of the policy on each task is $O(1/\sqrt{NT}) + O(\epsilon) + c$ for some constant $c$ (where suboptimality is defined according to Theorem 1).*

Recall that each task is formed by selecting a subset $\mathcal{U}_t$ of entities from the global set $\mathcal{U}$. We thus have the following sample gain that can be attributed to the permutation invariance of the policy network.

**Corollary 2 (Gain in Sample Efficiency from Permutation Invariance)** *Let $M = |\mathcal{U}|$ and $m = |\mathcal{U}_t|$. Given fixed $M$ and $m$, there are $T = \binom{M}{m} \geq \left(\frac{M}{m}\right)^m$ different tasks. Then, by Cor. 1, assuming all pairs of tasks are weakly correlated, the potential gain in sample efficiency is exponential in $m$.*

Disregarding correlation between samples from tasks with overlapping entities Corollary 1 and Corollary 2 together suggest that the (up to) exponential increase in the number of available tasks can significantly improve sample efficiency as compared to learning each task separately.

## 5 EXPLOITING PERMUTATION INVARIANCE THROUGH MULTI-TASK REINFORCEMENT LEARNING

Our approach to exploiting permutation invariance is via multi-task reinforcement learning, where each "task" corresponds to a particular choice of subset $\mathcal{U}_t \subset \mathcal{U}$. Furthermore, for each task, we enforce permutation invariance among the entities $i$ by forcing the neural network to apply the same sequence of operations to the state input $x_i$ of each instrument through parameter sharing.

The proposed method, shown in Algorithm 1, learns a single policy by sampling subsequences of trajectories from the different MDPs. At each step, we sample a task $t$ according to a distribution defined by task selection policy $p$. Then, a minibatch sample $\mathcal{B}_t$ is drawn from the replay buffer for task $t$, and gradient descent is performed using the sampled transitions $\mathcal{B}_t$ (alternatively, samples can be generated using policy rollouts for the specific task). Separate replay buffers maintained for each task are updated only when the corresponding task is being used.

In contrast with other active sampling approaches in multi-task learning, our approach maintains an estimate of the difficulty of each task $t$ as a score, $s_t$. After each training step, we update the score for only the sampled task based on minibatch $\mathcal{B}_t$, avoiding evaluation over all the tasks. The scoring functions depend on the sampled minibatch; to reduce fluctuations in scores for each task, exponential smoothing is applied $s_t \leftarrow \gamma s_t + (1 - \gamma) \cdot \mathrm{scorer}(\mathcal{B}_t)$. We propose a stochastic prioritization method that interpolates between pure greedy prioritization and uniform random sampling. Our approach is similar to prioritized experience replay (PER) by Schaul et al. (2016), but while classical PER prioritizes samples, we prioritize tasks. The probability of sampling task $t$ is $p_t = s_t^\alpha / \sum_{t'} s_{t'}^\alpha$, where the exponent $\alpha$ determines the degree of prioritization, with $\alpha = 0$ corresponding to the uniform case. We correct for bias with importance-sampling (IS) weights $w_t = 1/(Tp_t)^\beta$, that compensate for non-uniform probabilities if $\beta = 1$. We normalize weights by $1/\max_t w_t$. Tasks on which the reward variance is high can be interpreted as having more challenging samples, hence reward variance can be used as a scoring function.

---

**Algorithm 1** Prioritized Multi-Task Reinforcement Learning for Increasing Sample Efficiency

Initialize policy network $\pi_\theta$
Initialize replay buffers $R_1, \ldots, R_T$
Initialize time steps $n_1 \leftarrow 1, \ldots, n_T \leftarrow 1$
**loop**
    Select a task $t \sim p$ to train on
    Sample a random minibatch $\mathcal{B}_t$ of transitions $(x_n, a_n, r_n, x_{n+1})$ from $R_t$
    Update policy $\theta$ using $\mathcal{B}_t$ and chosen RL approach (correcting for bias using IS weights $w$)
    Update score $s_t \leftarrow \gamma s_t + (1 - \gamma) \cdot \mathrm{scorer}(\mathcal{B}_t)$
    Update ALL selection probabilities $p$ and IS weights $w$
    **for** $n = n_t, \ldots, \min\{n_t + n_e, N\}$ **do**
        For task $t$, select action $a_n$ according to current policy and exploration noise
        Execute action $a_n$, and observe reward $r_n$ and new state $x_{n+1}$
        Store transition $(x_n, a_n, r_n, x_{n+1})$ in $R_t$
    **end for**
    If $n < N$, update $n_t \leftarrow n + 1$, otherwise, update $n_t \leftarrow 1$
**end loop**

---

## 6 EXPERIMENTS

### 6.1 SYNTHETIC DATA

With the aim of validating the theory presented in Section 4, we define a synthetic example to explore the efficiency gain afforded by permutation invariance. To do so, we control of the deviation $\epsilon$ between any two tasks, thereby empirically validating the main theoretical results.

Consider a resource allocation problem where the observed state $x_i$ for each entity $i \in \{1 \ldots m\}$ is a single scalar $x_i \in [0, 1]$. The action space is the probability simplex, where each action

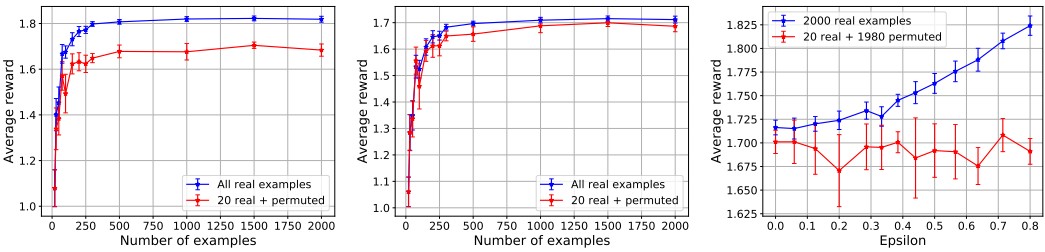

Figure 1: Performance for $\epsilon = 0.8$ (left), $\epsilon = 0$ (middle), and at $N = 2000$ with varying $\epsilon$ (right).

$a = (a_1 \ldots a_m)$ indicates the fraction of resource allocated to each entity. The reward function is

$$R(x, a) := \sum_i x_i a_i - \beta_i a_i \log a_i$$

where $\beta_i$ is a weight parameter for each entity. Note that when $\beta_i = \beta$ for all $i$, the reward function becomes $R(x, a) = (\sum_i x_i a_i) + \beta H(a)$ where $H$ is the Shannon entropy. This implies that maximizing the reward involves a tradeoff between focusing resources on high $x_i$ or distributing them uniformly across all $i$. Note that the reward function is permutation invariant, but that when we allow a varying $\beta_i$ over the entities, the function deviates from being perfectly permutation invariant. We use the range $\max_i \beta_i - \min_i \beta_i$ as a stand-in for $\epsilon$. Let $m = 10$. For each $\epsilon$, we run two experiments. The first examines the performance of policies trained by $LSPI$ using $N$ real examples drawn i.i.d from the state-action space, for $N = 20 \ldots 2000$. A small Gaussian noise is added to each reward to make learning harder. The second experiment uses only 20 real examples, but augments the training set (up to $N$) through random permutation of the real examples. The first two figures in Fig. 1 show the results for $\epsilon = 0.8$ and $\epsilon = 0$, respectively. Performance improves with $N$, as predicted by the $1/\sqrt{N}$ term in our error bound. Note that in the experiment using only 20 real examples, a performance gain is achieved by using permuted examples; this corresponds precisely to the multi-task gain predicted by the $1/\sqrt{NT}$ term. When $\epsilon$ is large, there is a significant gap between the results of the two experiments, as predicted by the $\epsilon$-term in the error bound. The last plot in Fig. 1 shows this gap at $N = 2000$ when $\epsilon$ varies from 0 to 0.8.

## 6.2 REAL-WORD DATA

We consider two real-world resource allocation settings: financial portfolio optimization and meta federated learning. Financial portfolio optimization is discussed below while meta federated learning is in the Appendix. Given historical prices for a universe of financial assets, $\mathcal{U}$, the goal of task $t$ is to allocate investments across a subset of assets $\mathcal{U}_t \subseteq \mathcal{U}$. The multiple tasks $t$ thus correspond to multiple portfolios of instruments. Permutation invariance will be of use in this setting since, from a given universe of instruments (e.g. the 500 instruments in the S&P 500), an exponential number of tasks can be generated, each with its own portfolio. Consider now one such task.

At the beginning of time period $n$, the action $a_{i,n}$ represents the fraction of wealth the decision maker allocates to asset $i$. The allocations evolve over the time period due to changes in asset prices. Let $w_{i,n}$ denote the allocation of asset $i$ at the end of time period $n$. We model the state of an asset using its current allocation and a window of its $H$ most recent prices. In particular, let $v_{i,n}$ denote the close price of asset $i$ over time period $n$, and let $y_{i,n} = v_{i,n}/v_{i,n-1}$ denote the ratio of close prices between adjacent time periods [1]. Then, the allocation in asset $i$ at the end of time period $n$ is given by

$$w_{i,n} = \frac{a_{i,n} y_{i,n}}{\sum_{i \in \mathcal{U}_t} a_{i,n} y_{i,n}},$$

and the state of asset $i$ at the beginning of time period $n$ is given by

$$x_{i,n} = (w_{i,n-1}, v_{i,n-H}/v_{i,n-1}, \ldots, v_{i,n-2}/v_{i,n-1}).$$

---

[1]Daily high and low prices are also used in the state but omitted here for brevity.

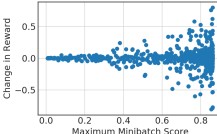 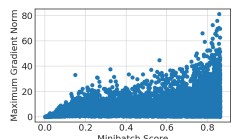 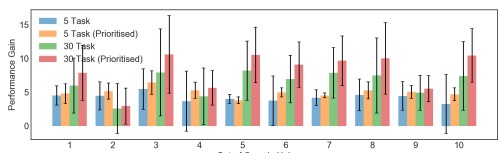

Figure 2: Scatter plots of the maximum absolute deviation from Equal CRP vs. the change in rewards every 50 steps (left) and the max. norm of the gradient for the minibatch (right).

Figure 3: Mean performance gain over Equal CRP of the learned policies when tested on 10 tasks using out-of-sample instruments. Error bars denote the standard deviation over 10 experiments.

The change in portfolio value over period $n$ depends on the asset prices and transaction costs incurred in rebalancing the portfolio from $(w_{i,n-1})_{i \in \mathcal{U}_t}$ to $(a_{i,n})_{i \in \mathcal{U}_t}$. The reward over period $n$ is defined as the log rate of return:

$$R_t(x_n, a_n) = \ln \left[ \beta \left( (w_{i,n-1})_{i \in \mathcal{U}_t}, (a_{i,n})_{i \in \mathcal{U}_t} \right) \sum_{i \in \mathcal{U}_t} a_{i,n} y_{i,n} \right],$$

where $\beta$ can be evaluated using an iterative procedure (see Jiang et al. (2017)). Defining the reward this way is appealing because maximizing average total reward over consecutive periods is equivalent to maximizing the total rate of return over the periods. To leverage this, we approximate $\beta((w_{i,n-1})_{i \in \mathcal{U}_t}, (a_{i,n})_{i \in \mathcal{U}_t}) \approx c \sum_{i \in \mathcal{U}_t} |w_{i,n-1} - a_{i,n}|$, where $c$ is a commission rate to obtain a closed-form expression for $R_t(x_n, a_n)$ (see Jiang et al. (2017)). We optimize using direct policy gradient on minibatches of consecutive samples

$$\theta \leftarrow \theta + \eta \nabla_\theta \left[ \frac{1}{B} \sum_{n=n_b}^{n_b+B-1} w_t R_t(x_n, \pi_\theta(x_n)) \right],$$

where $n_b$ is the first time index in the minibatch, $B$ the size of a minibatch, and $w_t$ the IS weight for task $t$. As in Jiang et al. (2017), we sample $n_b$ from a geometric distribution that prioritises recent samples and implement replay buffers for each task. A benchmark trading strategy is equal constantly-rebalanced portfolio (CRP) that rebalances to maintain equal weights. As we noted earlier, ideally one would prefer for the scoring function to depend only on the minibatch $\mathcal{B}_t$. A deviation from Equal CRP can be viewed as *learning to exploit price movements*, and is thus here we use this as the goal of the policy. *Prioritised MTL* thus prioritises tasks which deviate from Equal CRP. Note that the policy deviates from CRP only when profitable. Let

$$\mathsf{scorer}(\mathcal{B}_t) = \max_{n \in \{n_b, \ldots, n_b+B-1\}} \left\| \pi_\theta(x_n) - \frac{1}{|\mathcal{U}_t|} \right\|_\infty,$$

be the scoring of tasks in Prioritised MTL using mean absolute deviation of the minibatch allocation from Equal CRP. Figure 2 (left) shows a scatter plot of the maximum score seen every 50 steps and the change in episode rewards in a single-task learning experiment, and (right) of the minibatch score and the maximum gradient norm for the minibatch. Higher scores imply higher variance in the episode rewards and hence more challenging and useful samples. The correlation between scores and gradient norms shows that our approach is performing gradient-based prioritisation, (see Katharopoulos & Fleuret (2018); Loshchilov & Hutter (2015); Alain et al. (2015)) but in a computationally efficient manner. The details of the dataset and parameter settings can be found in the Appendix. Figure 3 shows the performance of the learned policies tested on 10 tasks drawn from out-of-sample instruments. The policy network with weights initialized close to zero behaves like an Equal CRP policy. As noted, any profitable deviation from Equal CRP implies learning useful trading strategies. The plots show that the MTL policies perform well on instruments never seen during training, offering a remarkable benefit for using RL in the design of trading policies. Fig. 4 shows the performance of prioritised multi-task learning (MTL) versus single-task learning (STL) (i.e. learning a policy for each task independently on the instruments in the task). We also show results for MTL without prioritised sampling, i.e., with $\alpha = 0$. We consider 5 tasks and 30 tasks. The plots show that prioritised MTL performs significantly better than STL in both convergence time and final achieved

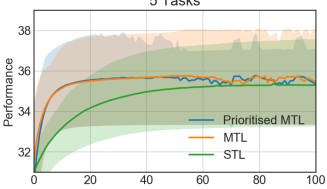 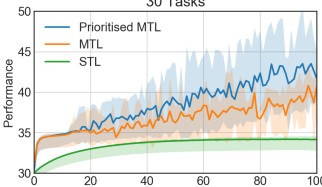 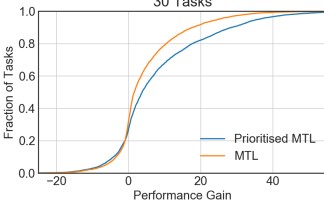

Figure 4: The two left plots show mean annualized return in the testing period over 10 experiments (different instruments) each with 5 and 30 tasks. X-axes are scaled to make the curves comparable: each epoch has 1500 (5-tasks) and 9000 steps (30-tasks) and an evaluation. Shaded regions denote the interquartile range. The rightmost figure shows, for each fraction of tasks, the gain over Single Task Learning (STL). A curve further to the right shows higher gain over STL. From 30% on the y-axis, the P-MTL gain is higher (more towards the right) than the MTL gain. As expected, when few tasks are used, prioritizing tasks doesn't help much (y-axis from 0 to 0.2).

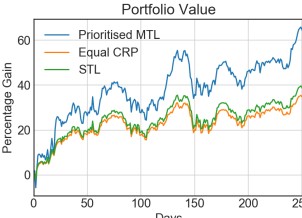 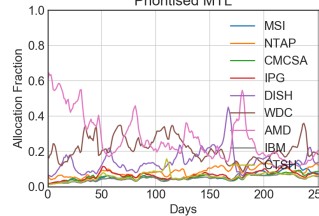 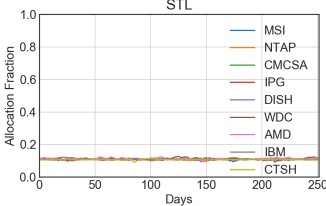

Figure 5: Comparison of a multi-task policy vs. a single-task policy on the testing period for a specific task. The leftmost plot shows the percentage gain in portfolio value over time for both policies against that from the baseline Equal CRP policy. The right two plots show the asset allocations.

performance. The performance with 30 tasks is significantly better than the performance with 5 tasks, showing that our approach leverages the samples of the additional tasks.

Fig. 5 illustrates the typical behavior of a multi-task learning (MTL) and a single-task learning (STL) policy on the test period for tasks where multi-task policy performed significantly better. The single-task policy kept constant equal allocations while the multi-task policy was able to learn more complex allocations. In financial data, strongly trending prices do not occur often and are inherently noisy. Multi-task learning with permutation invariance helps with both challenges, allowing the algorithm to learn more complex patterns in a given training period.

## 7 CONCLUSIONS

We introduce an approach for increasing the sample efficiency of reinforcement learning in a setting with widespread applicability within the class of sequential resource allocation problems. This property is permutation invariance: resources are allocated to entities according to a score, and the order can change without modifying the optimal allocation. Under this property, we show that a bound exists on the policy performance. This bound motivates a highly effective algorithm for improving the policy through a multi-task approach. Using prioritized task-sampling, the method not only improves the reward of the final policy but also renders it more robust. We illustrate the property and the method on two important problems: sequential financial portfolio optimization and meta federated learning, where the latter is provided in the Appendix.

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

## A    APPENDIX

**Theorem 1**    Let $\mathcal{M} = \langle \mathcal{X}, \mathcal{A}, R, P, \gamma \rangle$ be an MDP with reward function $R$ and transition kernel $P$. Denote its Bellman operator by

$$(\mathcal{T}^\pi V)(x) = R^\pi(x) + \gamma \int_{\mathcal{X}} P^\pi(dy|x)V(y).$$

Given a policy $\pi$, define the Bellman difference operator between $\mathcal{M}_t$ and $\mathcal{M}$ to be $\mathcal{D}_t^\pi V = \mathcal{T}_t^\pi V - \mathcal{T}^\pi V$. Apply the LSPI algorithm to $\mathcal{M}$, by generating, at each iteration $k$, a path from $\mathcal{M}$ of size $N$, where $n$ satisfies Lemma 4 in Antos et al. (2008). Let $V_{-1} \in \tilde{\mathcal{F}}$ be an arbitrary initial value function, $V_0, \cdots, V_{K-1}$ ($\tilde{V}_0, \cdots, \tilde{V}_{K-1}$) be the sequence of value functions (truncated value functions) generated by the LSPI after $K$ iterations, and $\pi_k$ be the greedy policy w.r.t. the truncated value function $\tilde{V}_{k-1}$. Suppose also that

$$\|\mathcal{D}_t^\pi V^\pi\|_\mu \leq \epsilon \, \forall \, \pi, \quad \text{and} \quad \|\mathcal{D}_t^{\pi_k} \tilde{V}_{k-1}\|_\mu \leq \epsilon \, \forall \, k.$$

Then, with probability $1 - \delta$ (with respect to the random samples), we have

$$
\begin{aligned}
\|V_t^{\pi_t^*} - V_t^{\pi_K}\|_\sigma \leq \frac{2}{(1-\gamma)^2} \Bigg\{ &(1+\gamma)\sqrt{CC''_{\sigma,\mu}} \Bigg[ \frac{2}{\sqrt{1-\gamma^2}} \left( 2\sqrt{2}E_0(\mathcal{F}) + E_2 \right) \\
&+ \frac{2}{1-\gamma} \left( \gamma V_{\max} L \sqrt{\frac{d}{\nu_\mu}} (\sqrt{\frac{8\log(8dK/\delta)}{N}} + \frac{1}{N}) \right) + E_1 \Bigg] \\
&+ \gamma^{\frac{K-1}{2}} R_{\max} + 3\epsilon\sqrt{2C'_{\sigma,\mu}} \Bigg\}.
\end{aligned}
$$

**Proof:** For convenience, we will simply remove the task subscript whenever we refer to variables associated with $\mathcal{M}$. Define

$$d_t^\pi = \mathcal{D}_t^\pi V^\pi,$$
$$\tilde{d}_{t,k} = \mathcal{D}_t^{\pi_k} \tilde{V}_{k-1},$$
$$e_k = \tilde{V}_k - \mathcal{T}^{\pi_k} \tilde{V}_k,$$
$$E_k = P^{\pi_{k+1}}(I - \gamma P^{\pi_{k+1}})^{-1} - P^{\pi^*}(I - \gamma P^{\pi_k})^{-1},$$
$$F_k = P^{\pi_{k+1}}(I - \gamma P^{\pi_{k+1}})^{-1} + P^{\pi^*}(I - \gamma P^{\pi_k})^{-1}.$$

From the proof of Lemma 12 in Antos et al. (2008), we get

$$V^{\pi^*} - V^{\pi_K} \leq \gamma \sum_{k=0}^{K-1} (\gamma P^{\pi^*})^{K-k-1} E_k e_k + (\gamma P^{\pi^*})^K (V^{\pi^*} - V^{\pi_0}).$$

By applying the above inequality, and taking the absolute value on both sides point-wise, we get

$$|V_t^{\pi_t^*} - V_t^{\pi_K}|$$
$$= |V_t^{\pi_t^*} - V^{\pi^*}| + |V^{\pi^*} - V^{\pi_K}| + |V^{\pi_K} - V_t^{\pi_K}|$$
$$\leq \gamma \sum_{k=0}^{K-1} (\gamma P^{\pi^*})^{K-k-1} F_k |e_k| + (\gamma P^{\pi^*})^K |V^{\pi^*} - V^{\pi_0}| + |V_t^{\pi_t^*} - V^{\pi^*}| + |V^{\pi_K} - V_t^{\pi_K}|$$
$$\leq \gamma \sum_{k=0}^{K-1} (\gamma P^{\pi^*})^{K-k-1} F_k |e_k| + \frac{2R_{\max}}{1-\gamma} \gamma^K + |V_t^{\pi_t^*} - V^{\pi^*}| + |V^{\pi_K} - V_t^{\pi_K}|$$

where we used the fact that $|V^{\pi^*} - V^{\pi_0}| \leq (2R_{\max}/(1-\gamma))\mathbf{1}$. Next, we derive upper bounds for $|V_t^{\pi_t^*} - V^{\pi^*}|$ and $|V^{\pi_K} - V_t^{\pi_K}|$.

(a) Observe that

$$V_t^{\pi_t^*} - V^{\pi^*} = \mathcal{T}_t^{\pi_t^*} V_t^{\pi_t^*} - \mathcal{T}^{\pi^*} V^{\pi^*}$$
$$\leq \mathcal{T}_t^{\pi_t^*} V_t^{\pi_t^*} - \mathcal{T}^{\pi_t^*} V^{\pi^*}$$
$$= \mathcal{T}_t^{\pi_t^*} V_t^{\pi_t^*} - \mathcal{T}^{\pi_t^*} V_t^{\pi_t^*} + \mathcal{T}^{\pi_t^*}(V_t^{\pi_t^*} - V^{\pi^*})$$
$$\leq (I - \gamma P_t^{\pi_t^*})^{-1} d_t^{\pi_t^*}.$$

The first inequality follows from the fact that $\pi^*$ is optimal with respect to $V^{\pi^*}$. The second inequality follows from the taylor expansion of the inverse term. By closely following the same steps, we also get

$$V_t^{\pi_t^*} - V^{\pi^*} = \mathcal{T}_t^{\pi_t^*} V_t^{\pi_t^*} - \mathcal{T}^{\pi^*} V^{\pi^*}$$
$$\geq \mathcal{T}_t^{\pi^*} V_t^{\pi_t^*} - \mathcal{T}^{\pi^*} V^{\pi^*}$$
$$= \mathcal{T}_t^{\pi^*} V^{\pi^*} - \mathcal{T}^{\pi^*} V^{\pi^*} + \mathcal{T}_t^{\pi^*}(V_t^{\pi_t^*} - V^{\pi^*})$$
$$\geq (I - \gamma P_t^{\pi^*})^{-1} d_t^{\pi^*}.$$

By splitting into positive and negative components and applying the above bounds, we get

$$|V_t^{\pi_t^*} - V^{\pi^*}| = |(V_t^{\pi_t^*} - V^{\pi^*})_+ - (V_t^{\pi_t^*} - V^{\pi^*})_-|$$
$$\leq |(V_t^{\pi_t^*} - V^{\pi^*})_+| + |(V_t^{\pi_t^*} - V^{\pi^*})_-|$$
$$\leq |(I - \gamma P_t^{\pi_t^*})^{-1} d_t^{\pi_t^*}| + |(I - \gamma P_t^{\pi^*})^{-1} d_t^{\pi^*}|$$
$$\leq (I - \gamma P_t^{\pi_t^*})^{-1} |d_t^{\pi_t^*}| + (I - \gamma P_t^{\pi^*})^{-1} |d_t^{\pi^*}|$$

(b) Observe that

$$V^{\pi_K} - V_t^{\pi_K} \leq \mathcal{T}^{\pi_K} V^{\pi_K} + \mathcal{T}^{\pi_K} \tilde{V}_{K-1} - \mathcal{T}_t^{\pi_K} \tilde{V}_{K-1} - \mathcal{T}_t^{\pi_K} V_t^{\pi_K}$$

$$= \mathcal{T}^{\pi_K} V^{\pi_K} + \mathcal{T}^{\pi_K} \tilde{V}_{K-1} - \mathcal{T}_t^{\pi_K} \tilde{V}_{K-1} - \mathcal{T}_t^{\pi_K} V^{\pi_K} + \mathcal{T}_t^{\pi_K} (V^{\pi_K} - V_t^{\pi_K})$$

$$\leq (I - \gamma P_t^{\pi_K})^{-1} (-d_t^{\pi_K} - \tilde{d}_{t,K}).$$

The first inequality follows from the fact that $\pi_K$ is optimal with respect to $\tilde{V}_{K-1}$. The second inequality follows from the taylor expansion of the inverse term. By closely following the same steps, we also get

$$V^{\pi_K} - V_t^{\pi_K} \geq \mathcal{T}^{\pi_K} V^{\pi_K} - \mathcal{T}^{\pi_K} \tilde{V}_{K-1} + \mathcal{T}_t^{\pi_K} \tilde{V}_{K-1} - \mathcal{T}_t^{\pi_K} V_t^{\pi_K}$$

$$= \mathcal{T}^{\pi_K} V^{\pi_K} - \mathcal{T}^{\pi_K} \tilde{V}_{K-1} + \mathcal{T}_t^{\pi_K} \tilde{V}_{K-1} - \mathcal{T}_t^{\pi_K} V^{\pi_K} + \mathcal{T}_t^{\pi_K} (V^{\pi_K} - V_t^{\pi_K})$$

$$\geq (I - \gamma P_t^{\pi_K})^{-1} (-d_t^{\pi_K} + \tilde{d}_{t,K}).$$

By splitting into positive and negative components and applying the above bounds, we get

$$|V^{\pi_K} - V_t^{\pi_K}| = |(V^{\pi_K} - V_t^{\pi_K})_+ - (V^{\pi_K} - V_t^{\pi_K})_-|$$

$$\leq |(V^{\pi_K} - V_t^{\pi_K})_+| + |(V^{\pi_K} - V_t^{\pi_K})_-|$$

$$= |(I - \gamma P_t^{\pi_K})^{-1} (-d_t^{\pi_K} - \tilde{d}_{t,K})| + |(I - \gamma P_t^{\pi_K})^{-1} (-d_t^{\pi_K} + \tilde{d}_{t,K})|$$

$$\leq (I - \gamma P_t^{\pi_K})^{-1} |-d_t^{\pi_K} - \tilde{d}_{t,K}| + (I - \gamma P_t^{\pi_K})^{-1} |-d_t^{\pi_K} + \tilde{d}_{t,K}|$$

$$\leq 2(I - \gamma P_t^{\pi_K})^{-1} (|d_t^{\pi_K}| + |\tilde{d}_{t,K}|).$$

By applying the upper bounds from (a) and (b), we get

$$|V_t^{\pi_t^*} - V_t^{\pi_K}| \leq \frac{2(1 - \gamma^{K+2})}{(1 - \gamma)^2} \left[ \sum_{k=0}^{K-1} \alpha_k A_k |e_k| + \alpha(R_{\max}/\gamma) \right.$$

$$+ (\beta/6) B^{\pi_t^*} \cdot 6|d_t^{\pi_t^*}| + (\beta/6) B^{\pi^*} \cdot 6|d_t^{\pi^*}|$$

$$\left. + (\beta/3) B^{\pi_K} \cdot 6|d_t^{\pi_K}| + (\beta/3) B^{\pi_K} \cdot 6|\tilde{d}_{t,K}| \right]$$

where we introduced the positive coefficients

$$\alpha_k = \frac{(1 - \gamma)}{1 - \gamma^{K+2}} \gamma^{K-k}, \quad \text{for } 0 \leq k < K,$$

$$\alpha = \frac{(1 - \gamma)}{1 - \gamma^{K+2}} \gamma^{K+1},$$

$$\beta = \frac{(1 - \gamma)}{2(1 - \gamma^{K+2})},$$

and the operators

$$A_k = \frac{1 - \gamma}{2} (P^{\pi^*})^{K-k-1} F_k, \quad \text{for } 0 \leq k < K,$$

$$B^{\pi} = (1 - \gamma)(I - \gamma P_t^{\pi})^{-1}.$$

Let $\lambda_K = \left[ \frac{2(1 - \gamma^{K+2})}{(1 - \gamma)^2} \right]^p$. Note that the coefficients $\alpha_k$, $\alpha$, and $\beta$, sum to 1, and the operators are positive linear operators that satisfy $A_k \mathbf{1} = \mathbf{1}$ and $B^{\pi} \mathbf{1} = \mathbf{1}$. Therefore, by taking the $p$th power on both sides, applying Jensen's inequality twice, and then integrating both sides with respect to $\sigma(x)$, we get

$$\|V_t^{\pi_t^*} - V_t^{\pi_K}\|_{p,\sigma}^p = \int \sigma(dx) |V_t^{\pi_t^*} - V_t^{\pi_K}|^p$$

$$\leq \lambda_K \sigma \left[ \sum_{k=0}^{K-1} \alpha_k A_k |e_k|^p + \alpha(R_{\max}/\gamma)^p \right.$$

$$+ (\beta/6) B^{\pi_t^*} (6|d_t^{\pi_t^*}|)^p + (\beta/6) B^{\pi^*} (6|d_t^{\pi^*}|)^p$$

$$\left. + (\beta/3) B^{\pi_K} (6|d_t^{\pi_K}|)^p + (\beta/3) B^{\pi_K} (6|\tilde{d}_{t,K}|)^p \right].$$

From the definition of the coefficients $c_{\sigma,\mu}(m)$, we get

$$\sigma A_k \leq (1-\gamma) \sum_{m \geq 0} \gamma^m c_{\sigma,\mu}(m + K - k)\mu,$$

$$\sigma B^\pi \leq (1-\gamma) \sum_{m \geq 0} \gamma^m c_{\sigma,\mu}(m)\mu.$$

Therefore, it follows that

$$\sigma \left[ \sum_{k=0}^{K-1} \alpha_k A_k |e_k|^p \right] \leq (1-\gamma) \sum_{k=0}^{K-1} \alpha_k \sum_{m \geq 0} \gamma^m c_{\sigma,\mu}(m + K - k)\mu|e_k|^p$$

$$= \frac{\gamma(1-\gamma)^2}{1-\gamma^{K+2}} \sum_{k=0}^{K-1} \sum_{m \geq 0} \gamma^{m+K-k-1} c_{\sigma,\mu}(m + K - k)\|e_k\|_{p,\mu}^p$$

$$\leq \frac{\gamma}{1-\gamma^{K+2}} C''_{\sigma,\mu} e^p$$

where $e = \max_{0 \leq k < K} \|e_k\|_{p,\mu}^p$. The terms involving $B^\pi$ satisfy

$$\sigma \left[ B^\pi (6|d_t^\pi|)^p \right] \leq 6^p (1-\gamma) \sum_{m \geq 0} \gamma^m c_{\sigma,\mu}(m)\mu|d_t^\pi|^p \leq 6^p C'_{\sigma,\mu} \|d_t^\pi\|_{p,\mu}^p.$$

Putting all these together, and choosing $p = 2$, we get

$$\|V_t^{\pi_t^*} - V_t^{\pi_K}\|_\sigma \leq \lambda_K^{\frac{1}{2}} \left[ \frac{\gamma}{1-\gamma^{K+2}} C''_{\sigma,\mu} e^2 + \frac{(1-\gamma)\gamma^{K+1}}{1-\gamma^{K+2}} (R_{\max}/\gamma)^2 + \frac{36(1-\gamma)}{2(1-\gamma^{K+2})} C'_{\sigma,\mu} \epsilon^2 \right]^{\frac{1}{2}}$$

$$\leq \frac{2}{(1-\gamma)^2} \left[ \gamma C''_{\sigma,\mu} e^2 + (1-\gamma)\gamma^{K+1} (R_{\max}/\gamma)^2 + \frac{36(1-\gamma)}{2} C'_{\sigma,\mu} \epsilon^2 \right]^{\frac{1}{2}}$$

$$\leq \frac{2}{(1-\gamma)^2} \left[ C''_{\sigma,\mu} e^2 + \gamma^{K+1} (R_{\max}/\gamma)^2 + 18 C'_{\sigma,\mu} \epsilon^2 \right]^{\frac{1}{2}}$$

$$\leq \frac{2}{(1-\gamma)^2} \left[ \sqrt{C''_{\sigma,\mu}} e + \gamma^{\frac{K-1}{2}} R_{\max} + 3\epsilon \sqrt{2 C'_{\sigma,\mu}} \right].$$

The desired result can then be obtained by applying the same steps as in the proof of Theorem 8 in Lazaric et al. (2012).

$\square$

## A.1 FINANCIAL PORTFOLIO OPTIMIZATION: ADDITIONAL DETAILS

The dataset consists of daily prices for 68 instruments in the technology and communication sectors from 2009 to 2019. We use 2009–2018 for training and 2019 for testing. To validate that our approach learns common features across instruments, and thus can *transfer*, we reserve 18 instruments not seen during training for further testing. The global asset universe $\mathcal{U}$ used for training contains 50 instruments.

We construct tasks by randomly choosing a portfolio of $|\mathcal{U}_t| = 10$ instruments for each task. We create a permutation invariant policy network by applying the same sequence of operations to every instrument state. That is, for each instrument, the flattened input prices are passed through a common RNN with 25 hidden units and tanh activation, this output is concatenated with the latest allocation fraction of the instrument, and passed through a common dense layer to produce a score. Instrument scores are passed to a softmax function to produce allocations that sum to one. The smoothing parameter for the scores $\gamma = 0.2$, $\alpha = 0.5$ for the task prioritisation parameter and $\beta = 1.0$ to fully compensate for the prioritized sampling bias.

## A.2 META FEDERATED LEARNING

Suppose we have a universe of federated learning clients $\mathcal{U}$. The goal of task $t$ is to aggregate models in a federated learning experiment over a subset of clients $\mathcal{U}_t \subseteq \mathcal{U}$. At each step $n$, the action $a_{i,n}$

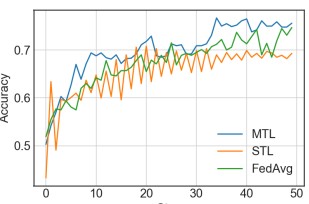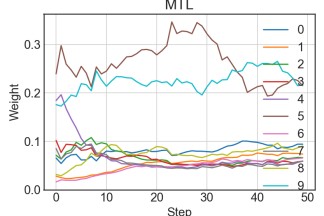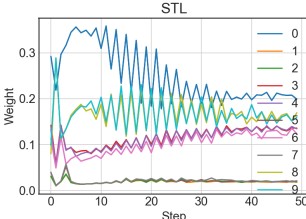

Figure 6: These plots compare the behavior of a multi-task policy and a single-task policy during testing. FedAvg denotes the accuracy of federated learning with uniform averaging. The left plot shows the accuracy of the aggregate model during federated learning. The right two plots show the weights produced by the policy for different clients. Note that clients 8 and 9 possess 40% of the unique labels.

represents the weight assigned to the supervised learning model of client $i$ in the averaging procedure. Let $v_{i,n}$ denote the model of the client (i.e. the tensor of model parameters). We model the state of the client as some function of its $H$ most recent models $x_{i,n} = f(v_{i,n-H+1}, \ldots, v_{i,n})$. Assume that the aggregator has access to a small evaluation dataset that it can use to approximately assess the quality of models. We define the reward at each step to be the accuracy of the aggregate model, $R_t(x_n, a_n) = \mathcal{L}\left(\sum_{i \in \mathcal{U}_t} a_{i,n} v_{i,n}\right)$, where $\mathcal{L}(v)$ is a function that provides the accuracy of a model $v$ on the evaluation dataset. Therefore, by maximizing the total return over all time periods, we seek to maximize both the accuracy at the final time step as well as the time to convergence. We optimize the policy using Proximal Policy Optimization (PPO).

We use the MNIST digit recognition problem. Each client observes 600 samples from the train dataset and trains a classifier composed of one 5x5 convolutional layer (with 32 channels and ReLu activation) and a softmax output layer. We use the same permutation invariant policy network architecture as before with 10 hidden units in the RNN. We randomly select $|\mathcal{U}_t| = 10$ clients for each task. We learn using an evaluation dataset comprised of 1000 random samples from the test dataset and test using all 10000 samples in the test dataset. We fix the number of federated learning iterations to 50.

We explore the benefit of MTL in identifying useful clients in scenarios with skewed data distribution. We partition the dataset such that 8 of the clients in each task observe random digits between 0 to 5 and the remaining 2 clients observe random digits between 6 to 9. Therefore, for each task, 20% of the clients possess 40% of the unique labels. The state of each client are the accuracies of its $H$ most recent models on the evaluation dataset.

Figure 6 shows the potential benefits of multi-task learning when simulators are inaccurate. In particular, we obtain two aggregation policies, one trained using single-task learning (STL), and another trained using multi-task learning (MTL), both trained using the same number of steps, and we observe their behavior during testing. The plots show that multi-task learning is able to learn non-uniform averaging policies that improve the convergence and performance of federated learning runs. More importantly, it can perform better than single-task learning even with the same number of samples. This may be attributed to the wider variety of client configurations (and consequently experiences) in the multi-task approach.

