# OpenReview forum: "Efficient Reinforcement Learning in Resource Allocation Problems Through Permutation Invariant Multi-task Learning"
_ICLR.cc/2021/Conference — Reject_

### Official Review · AnonReviewer4 · 2020-10-26
**A theoretically guaranteed multi-task reinforcement learning approach that addresses sequential resource allocation problems**

**Rating:** 7
**Confidence:** 4

**Review:**

This paper addresses sequential resource allocation problem using reinforcement learning, where sample efficiency is the focus of the paper. The authors identify a key property in the targeted resource allocation problems -- the permutation invariance -- which intrinsically implies the independency of samples at different time steps. From this property the paper extends the work of D’Eramo et al. (2020) and derives a potentially tighter bound for the gap between the optimal policy and the policy learned from multiple tasks. The paper also designs a new algorithms that prioritizes sampling in multi-task learning that addresses the bias between training and target tasks. Empirical evaluations on financial portfolio allocation and meta federated learning demonstrate the effectiveness of the proposed approach.

Strengths:
1) Using RL to solve sequential resource allocation problems is interesting and well-motivated, it can promote the impact of RL approaches when deployed
2) The paper has technical depth, and provides a theoretical guarantee of their approach
3) The paper gives a good discussion of existing works and where the paper lies in the line;
4) The empirical evaluations show two interesting case studies of sequential resource allocation problems. The paper contributes non-trivial efforts in designing experiment strategies for the two cases. The results demonstrate the effectiveness of the proposed approach.

Weakness:
A major weakness in my opinion is that the theorem is derived assuming the action space is finite, while in experiments and mathematical formulations the actions are denoted as continuous (thus infinite). This is however already clearly pointed out in the paper and placed as future work, so this is somewhat fine to me.

Some comments:
1) The paper gives a good discussion of existing works and where the paper lies in the line; it would be better if the authors can briefly discuss meta-RL which is close to the problem being studied in this paper
2) The significance of the theorem is well-discussed. However it is hard to understand what does Assumption 1 imply -- Is it a strong or weak assumption? At what circumstances does this assumption hold?
3) In the proof of the theorem, it is not clear -- what is the high-level intuition of the proof? How is permutation invariance property used?
4) Figure 1 is placed before 2-4 but referenced only in the end of the experiment section
5) Figure 2, the annotations left and right in the brackets are reversed.

Questions: please refer to Some comments points 2) and 3)

---

> ### Author Response · Authors · 2020-11-20
> **Response to Reviewer 4**
>
> We wish to thank Reviewer 4 for the constructive comments and suggestions. The paper has been revised and uploaded. The comments of the reviewer and corresponding changes made are summarized here.
>
> 1) Meta reinforcement learning. We have discussed meta learning in general in the Introduction and Related Work sections.
>
> 2) Interpretation of Assumption 1. Assumption 1 deals with properties on the controllability of an MDP. It provides the technical conditions necessary for deriving our finite-sample performance bounds. In particular, it requires that the two “concentrability” measures are finite; while this is not guaranteed in general, it is not a strong requirement. A more detailed discussion on this can be found in [Antos et al 2008].
>
> 3) Figure 1 placement (now called Figure 3 after adding the synthetic experiment). Thanks for bringing this up. The placement of figures and corresponding text has now been adjusted to flow better.
>
> 4) Figure 2 brackets (now moved to new position as per point 3 above, and still called Figure 2 after adding the synthetic experiment). Typo corrected, thank you!

---

### Official Review · AnonReviewer3 · 2020-10-28
**hard to read and understand**

**Rating:** 5
**Confidence:** 2

**Review:**

The paper proposes an RL algorithm for multi-task learning. Under certain assumptions, the paper proves a sample complexity result for this setting. The paper presents a new algorithm based on this approach. The empirical results on the task of sequential portfolio optimization shows that this approach performs better than the policy of constantly rebalanced portfolio.

I dont understand the aim of this paper and unfortunately, the paper did not help me either. It seems that the paper focuses on multi-task learning using RL. The main assumption the paper makes is permutation invariance (PI). However, the way paper defines it is not clear to me.

Def 1: "A policy network is PI if it satisfies pi(sigma(a), sigma(x)) = sigma(pi(a,x))" for any permutation sigma.
So sigma is a permutation of items in a set?
In this case it seems the set is the output of policy network. What is the output of policy network: action? But the action is not defined as a set of items. It seems that one must guess that the actions are division of 1 resource among t entities. Lets say t = 2 and resulting action is [0.2, 0.8]
If this is true the the two permutation of a are [0.2, 0.8] and [0.8, 0.2]. But then the definition says the left-hand side should be equal to both these permutations. So [0.2, 0.8] == [0.8, 0.2]?
I dont understand this.

By the defintion do you mean: pi(sigma(a), sigma(x)) = pi(a,x). This would mean that the rearrangement of state or actions do not change the output of the policy network. I can understand this but not the def in the paper.

Finally, what does this PI means in real-life? Since the paper talks about resource allocation, does the PI mean that it does not matter which entity the resource is being allocated to as long as the share of resource does not change? For what kind of problems does such an assumption/property hold? What is one entity is better at utilizing resources than others?

After going about the definition/assumption for quite some time, the main theorem of the paper does not even use the definition/assumption. So why was it introduced?

From related work "Lazaric et.al. provide a performance bound that bears similarity to ours, which one can consider an extension for our particular PI setting". So which approach is more general, the one in Lazaric et. al or yours. Seems yours since Lazaric et.al. is an extension to your PI setting. But then the same sentence says yours is a particular setting?

Corollary 2: " the gain in sample efficiency can be exponential in m". What is the meaning of 'can be'. Is it exponential or not? If yes, then under what condition?

How is portfolio optimization a multi-task RL problem? It can be formulated as a resource allocation problem but apart from maximizing long-term returns what are the other task the agent must perform for this problem.

Sorry to say that I tried quite hard but I could not make sense of either the text or mathematics of the paper.  It seems the paper presents an interesting attempt at resource allocation using RL but I found the writing highly confusing.

---

> ### Author Response · Authors · 2020-11-20
> **Response to Reviewer 3**
>
> We wish to thank Reviewer 3 for the constructive comments and suggestions. The paper has been revised and uploaded. The comments of the reviewer and corresponding changes made are summarized here.
>
> 1) Clarify Definition 1.  There is a typo in our original Definition 1, where (a,x) is used as the state (due to the fact that we included the previous action as part of the current state). We apologize for the confusion caused and thank the reviewer for pointing this out. This has now been fixed. Both the state x and action a consist of components x=(x_1…,x_m) and a=(a_1,…,a_m), each corresponding to an entity. Sigma is a permutation function on these entities, while pi maps a state x to action a. Thus, being permutation invariant means that pi is not sensitive to the ordering (hence the identity) of the entities within x and a.
>
> 2) Theorem doesn’t use Definition/Assumption. The intended flow is as follows: Theorem 1 provides a general result and performance guarantee with respect to using data from a different but similar MDP. Definition 1 provides a basis for generating many such MDPs. Then, the benefit of doing so is demonstrated in Corollary 2. This explanation has been added to the paper before introducing the theoretical results. We thank the reviewer for insisting on this and we believe that the explanation added helps readers better understand the use of the theory.
>
> 3) More general, Lazaric vs. the present theorem. Our work extends the performance bounds on single-task RL of Lazaric et al 2012, and is indeed therefore more general. We have clarified this in the main text.
>
> 4) Corollary 2 language. The statement has been modified accordingly.
>
> 5) Hard to read and understand / How is portfolio optimization a multi-task problem?
> We have added a motivating example based on the financial portfolio optimization problem as paragraph 3 in the Introduction, (as suggested also by Reviewer 1).
> Regarding  how the portfolio optimization problem is a multi-task problem, this is now described in the above-mentioned motivating example of the Introduction as well as in the first paragraph of Section 6.2. We believe that, together, these additions help to clarify how the property of permutation invariance exists in resource allocation problems such as the financial application.
> In addition as per a suggestion by Reviewer 2, we have added a synthetic example in the beginning of the Experiment section that demonstrates both the value of the theory as well as the performance gain that the permutation invariance property provides.
>
> We hope that these additions along with the clarification of the use and flow of the theory (described in point 2, above) improve substantially the readability and clarity of the paper, and we thank the reviewer for the helpful suggestions.

---

### Official Review · AnonReviewer1 · 2020-10-29
**Interesting idea, adding motivating examples early will help**

**Rating:** 5
**Confidence:** 2

**Review:**

This paper addresses the problem of reinforcement learning using limited training samples. They propose a solution by exploiting the invariance property in the tasks. In particular, they present an algorithm that exploits permutation invariance, study its theoretical properties, and propose examples where this property holds and their algorithm can be leveraged.

I feel the paper could have been better presented by starting with a motivating example where the permutation invariance property holds - for example the portfolio optimization example studied in the experiments. This will make it easy to follow the multiple terminologies of tasks, entities, resources, introduced in Sec 3.

The setting considered in the paper is one where the state is a concatenation of various entities, while the actions are the fraction of resources allocated to each entity. The permutation invariance property is defined in Def. 1.

I didi not understand how a network trained using gradient descent alone would satisfy permutation invariance. There is no part of the pseudo code in Alg 1 explicitly making sure that the algorithm is permutation invariant.

---

> ### Author Response · Authors · 2020-11-20
> **Response to Reviewer 1**
>
> We wish to thank Reviewer 1 for the constructive comments and suggestions. The paper has been revised and uploaded. The comments of the reviewer and corresponding changes made are summarized here.
>
> 1) Start with a motivating example. We have added a motivating example based on the financial portfolio optimization problem, as suggested by Reviewer 1, as  paragraph 3 in the Introduction. This also helps to clarify (e.g. for Reviewer 3) how the portfolio optimization problem is a multi-task problem, and we thank Reviewer 1 for making a point of this, as we believe it helps improve the overall understanding of the paper.
>
> 2) Enforcing permutation invariance. Our approach enforces permutation invariance through parameter sharing in the policy network itself and not the algorithm. Thank you for suggesting that we better explain this. We have clarified it in Section 5. Additional information can also be found in the Appendix.

---

### Official Review · AnonReviewer2 · 2020-10-30
**good idea, presentation not convincing**

**Rating:** 5
**Confidence:** 3

**Review:**

This paper proposes an approach to reducing the sample complexity in multi-task reinforcement learning using permutation invariant policies. The main premise of the paper is that certain families of tasks exhibit approximate forms of symmetry, i.e., applying a permutation to the state/action variables would make all tasks similar in some metric sense. Then, the proposal is to learn a single permutation-invariant policy which will perform well on all of them simultaneously. An reinforcement learning algorithm to learn a permutation invariant policy is derived.

To get the technicalities out of the way, I think the paper is fairly well written and the related work is sufficiently well summarized (to my understanding). The authors are very open about the similarities to previous work and sources of inspiration. I checked the theory superficially and the results generally have the form I would expect (in terms of the involved quantities and their proportions), however, if there were more subtle issues, I certainly missed them.
A few minor nits: in the statement of theorem 1, the authors present important quantities in the bound as if they were universal constants (c1, c2, c3). These quantities are not problem independent, they involve features of the problem so the statement of thm 1 creates the wrong impression. Moreover the appendix refers to theorem 1 as "theorem 2", which was initially confusing to me. Also it seems that the symbol pi switches semantics a few times - first, it denotes a deterministic policy, then a stochastic policy, and finally a policy network. It may be good to disambiguate them somehow for the sake of readability.

I generally agree with the story, however, there are two major issues. First, the presentation is structured in a way that does not sell this work very strongly. A lot of effort is spent on setting up Theorem 1 in a completely detached way, and then it's actual relevance to the problem at hand is summarized in a hand-wavy corollary. From a paper claiming to solve resource allocation problems efficiently, I would expect at least the following: clearly articulate the problem. Argue convincingly that this particular problem class is hard (the arguments that N may be small and the simulator may be inaccurate are true for many problem classes). Explain the solution and how it exploits the structure of the problem. Finally present theory verifying the soundness of the solution.

The second issue concerns the application. While it's nice that the portfolio problem is somewhat grounded in reality, it's far from obvious that the theory explains what's going on in this problem. This problem is technically a POMDP, which is also suggested by the choice of on RNN as a policy network. The policy is then trained by policy gradient, which might be expected to behave differently on these problems. This gives us very little information about how good the proposed bound is. I believe this paper would benefit from a synthetic application where the difference in Bellman operators can be controlled precisely to see that the problem behaves as predicted by the theory.

Overall, I think that the general direction is good, and significant progress has been made, however, the current state of the paper does not present a convincing story.

---

> ### Author Response · Authors · 2020-11-20
> **Response to Reviewer 2**
>
> We wish to thank Reviewer 2 for the constructive comments and suggestions. The paper has been revised and uploaded. The comments of the reviewer and corresponding changes made are summarized here.
>
> 1) Theorem 1 constants. It has been noted that these constants are task and problem-specific in the text when they are introduced. We also made this explicit in Theorem 1.
>
> 2) Theorem numbering (main text vs. Appendix). The number 1 has been maintained in the Appendix to avoid ambiguity.
>
> 3) Meaning of pi. Although our approach does not require it, we have used deterministic policies throughout this work. We have updated the notation to make this clearer.
>
> 4) Presentation of the method. Better set up the problem and solution. The overall purpose of the method and indeed of the paper has been added as a last paragraph in the Introduction, and added at various other points throughout. In addition, as per a similar comment from Reviewer 3, the flow and use of the theory has been clarified before introducing the theorem in Section 4. Specifically, we added the explanation that Theorem 1 provides a general result and performance guarantee with respect to using data from a different but similar MDP. Definition 1 provides a basis for generating many such MDPs, and the benefit to be gained by doing so is demonstrated in Corollary 2.
>
> 5) Synthetic experiment to illustrate benefits of permutation invariance and significance of the theorem: This has been done and added to the beginning of the Experiment section. We believe that the addition of the synthetic example helps substantially to convey the message of the paper and convince readers of the benefit of the theory, and we are grateful to Reviewer 2 for this suggestion.

---

### Decision · Program_Chairs · 2021-01-07
**Final Decision**

**Decision:**

Reject

**Comment:**

This paper tackles a problem of resource allocation using reinforcement learning. An important invariant - permutation invariant - is identified as an important characteristic of this problem. Then it is shown that taking advantage of such an invariant should  dramatically improve the sample efficiency.

On behalf of the reviewers, I would like to thank the authors for addressing many concerns raised in the initial reviews. Unfortunately, a further examination revealed several other potential issues that require further clarification:

1. It seems that real-data experiments do not really demonstrate whether the benefits of the approach come from multi-task learning or from permutation invariance. It would make sense to run an ablation study. In particular, if the benefit is really coming from multi-task learning, then the theory part of the paper becomes less relevant.

2. The metric used for finance application appear to be in-adequate. It is typical in finance academic literature to look some form of risk-adjusted returns. Is the MTL strategy just taking more risk? How statistically significant are the results?

Given these concerns the paper can not be accepted in its current form but we encourage authors to address these and resubmit.